# Gene regulatory patterning codes in early cell fate specification of the *C. elegans* embryo

Alison G Cole[1,2†], Tamar Hashimshony[3†], Zhuo Du[4], Itai Yanai[5]*

[1]Department of Molecular Evolution and Development, University of Vienna, Vienna, Austria; [2]University of Vienna, Vienna, Austria; [3]Department of Biology, Technion – Israel Institute of Technology, Haifa, Israel; [4]State Key Laboratory of Molecular Developmental Biology, Institute of Genetics and Developmental Biology, Chinese Academy of Sciences, Beijing, China; [5]Institute for Computational Medicine, NYU School of Medicine, New York, United States

**Abstract** Pattern formation originates during embryogenesis by a series of symmetry-breaking steps throughout an expanding cell lineage. In *Drosophila*, classic work has shown that segmentation in the embryo is established by morphogens within a syncytium, and the subsequent action of the gap, pair-rule, and segment polarity genes. This classic model however does not translate directly to species that lack a syncytium – such as *Caenorhabditis elegans* – where cell fate is specified by cell-autonomous cell lineage programs and their inter-signaling. Previous single-cell RNA-Seq studies in *C. elegans* have analyzed cells from a mixed suspension of cells from many embryos to study late differentiation stages, or individual early stage embryos to study early gene expression in the embryo. To study the intermediate stages of early and late gastrulation (28- to 102-cells stages) missed by these approaches, here we determine the transcriptomes of the 1- to 102-cell stage to identify 119 embryonic cell states during cell fate specification, including 'equivalence-group' cell identities. We find that gene expression programs are modular according to the sub-cell lineages, each establishing a set of stripes by combinations of transcription factor gene expression across the anterior-posterior axis. In particular, expression of the homeodomain genes establishes a comprehensive lineage-specific positioning system throughout the embryo beginning at the 28-cell stage. Moreover, we find that genes that segment the entire embryo in *Drosophila* have orthologs in *C. elegans* that exhibit sub-lineage-specific expression. These results suggest that the *C. elegans* embryo is patterned by a juxtaposition of distinct lineage-specific gene regulatory programs each with a unique encoding of cell location and fate. This use of homologous gene regulatory patterning codes suggests a deep homology of cell fate specification programs across diverse modes of development.

**\*For correspondence:**
Itai.Yanai@nyulangone.org

†These authors contributed equally to this work

**Competing interest:** The authors declare that no competing interests exist.

## eLife assessment

This **valuable** work fills a gap in the mapping of gene expression patterns in the early embryo of *C. elegans*. The presented data are **solid** and provides a resource for future analysis.

## Introduction

Developmental toolkit genes constitute the genomic repertoire that underlies the pathways patterning the organism (*Carroll, 2008*). These genes are generally involved in signaling transduction modules and transcription factors (TFs) (*Degnan et al., 2009*; *Gerhart, 1999*). Their assembly into specific

temporal and spatial gene regulatory pathways leads to the specification of cell fates in the embryo. For example, the *Caenorhabditis elegans* endoderm is specified by a cascade of TFs operating sequentially (*Maduro and Rothman, 2002*). Identifying the principles of gene regulation has been a central endeavor in developmental biology.

John Sulston and colleagues elucidated the 1340 cells generated throughout the embryogenesis of the nematode *C. elegans*, leading to a hatched worm with 558 cells, each with a described name, lineage, and fate (*Sulston et al., 1983*). The cell lineage is delineated into a germ-line lineage – P – and five 'founder' somatic cell lineages – AB, MS, E, C, and D – that each have characteristic cell cycle timings (*Deppe et al., 1978*) and cell fates (*Figure 1a*). All founder cell lineages are present by the 28-cell stage, when gastrulation initiates, and by the 102-cell stage the progeny of most of the cells have a single fate.

The invariant cell lineage of the *C. elegans* embryo provides an opportunity to systematically study developmental gene pathways. Overall, there are 934 annotated TFs in *C. elegans*, with 40% having experimentally studied DNA-binding specificities (*Narasimhan et al., 2015*). The overall set can be classified into families according to their DNA-binding domain, including the homeodomain, bHLH, bZIP, T-box, C2H2 zinc finger, and the very large nuclear hormone receptor domain family. While mapping the location of signaling pathways is difficult from gene expression data, effects of signaling on the transcriptional output are detectable and consequently studying TF expression provides an entry point into gene expression pathway characterization.

Here, we use single-cell RNA-Seq to identify the transcriptomes of each of the cells up to the 102-cell stage. We find that homeodomain genes are expressed in stripes along the anterior-posterior axis, as early as the 28-cell stage. Interestingly, each founder cell lineage – AB, MS, C, and E – establishes its own regionalization code. These *Drosophila*-like stripe patterns suggest a deep homology in cell fate specification between the embryogenesis of non-segmented and segmented animals, despite differences in syncytium-based and cell-cleavage-based modes of development.

## Results

Previous studies analyzed cells from a mixed suspension of cells from different *C. elegans* embryos (*Packer et al., 2019*). However, this approach does not ensure sampling of each of the individual cells of the early stages, where it is difficult to dissociate the cells. A complementary approach is to manually collect cells by mouth pipette thus ensuring that all cells are collected. Other works employed this approach to study the early stages, however only arrived at the 15-cell stage (*Hashimshony et al., 2016*; *Tintori et al., 2016*). Up to this stage, few new TFs are induced and the transcriptome is dominated by the maternal deposit. As we were interested in studying early specification events in the embryo, we thus manually isolated individual cells from dissociated embryos and processed them for scRNA-Seq (*Figure 1b*). Overall, we studied 840 cells from 38 embryos up to the 102-cell stage, collecting all or most cells of each embryo (*Figure 1b* and *Supplementary file 1*).

Analyzing the transcriptomes, we found that embryo-to-embryo variation could be efficiently normalized by standardizing each gene's expression across all cells collected from the same embryo (see Methods), and this is another benefit of the manual collection approach. A dimensional reduction map of the examined cells (*Figure 1b*) reveals the unfolding of development with cells from the early stages occupying the center while cells from later stages occupy the periphery. We found that cells followed developmental trajectories according to their founder cell origin, each identified by the expression of genes known to be expressed in a lineage-specific manner: *ceh-51* (MS), *elt-7* (E), *pal-1* (C,D and P), *pes-1* (D), and *nos-2* (P) (*Figure 1c*; *Broitman-Maduro et al., 2009*; *Hunter and Kenyon, 1996*; *Lee et al., 2017*; *Maduro and Rothman, 2002*). As also observed by other studies (*Packer et al., 2019*), the founder cell lineage trajectories are oriented according to cell fates, with similar cell fates converging between lineages (*Figure 1b and d*): the cells of the ecto-mesoderm producing C lineage formed a bi-lobed cloud with the muscle part adjacent to that of the MS and D lineages. Similarly, MS cells that will form the pharynx orient toward the pharynx producing portion of the AB lineage.

In order to infer the precise identity of each cell, we first organized the embryos into eight developmental stages: the 1-, 2-, 4-, 8-, 15-, 28-, 51-, and 102-cell stages (*Figure 1a*). For each stage, we studied cells from multiple embryos and identified clusters of cells according to differential gene expression. *Figure 2a* demonstrates this analysis for the 15-cell stage and *Figure 2—figure supplement 1* for all

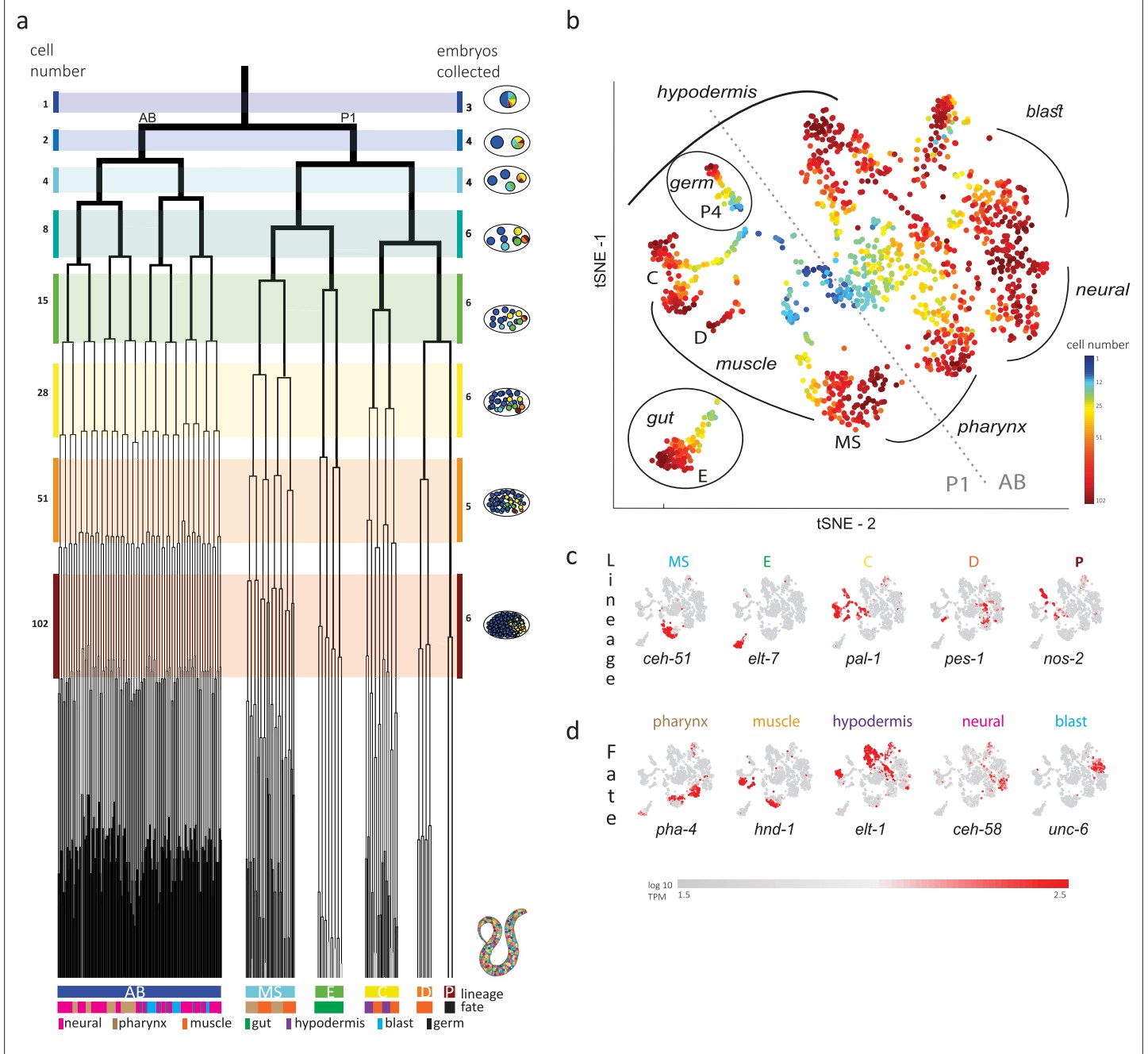

**Figure 1.** Single-cell transcriptomics in the early *C. elegans* embryo. (**a**) The embryonic *C. elegans* cell lineage as deciphered by *Sulston et al., 1983*. The number of cells present at each stage is indicated on the left and schematic images of the embryo are shown for several stages to the right with cells colored by founder cell lineages AB, MS, E, C, D, and P4. The fate of the cells at the last stage is also indicated according to the legend at the foot of the lineage. The number of embryos examined in this study is indicated on the right. Colored bars correspond to the windows in which embryos were considered to be at the same stage. (**b**) A t-distributed stochastic neighbor embedding (tSNE) map of the cells isolated in this study. Circles indicate the individual cells, colored by the stage of development of the embryo from which they were manually collected. (**c–d**) Gene expression for the indicated lineage and fate marker genes on the tSNE map shown in (**b**).

other stages. The identified clusters contain cells from each of the individual embryos, thus confirming our underlying assumption that embryo to embryo gene expression variation is minimal relative to the coherence of the cell states within an embryo. Finally, we inferred the cell identity of each cluster with reference to known gene markers from the literature (*Supplementary files 2 and 3*). *Figure 2b* illustrates our approach for the cells of the 28-cell stage (see *Figure 2—figure supplements 2–7* for all

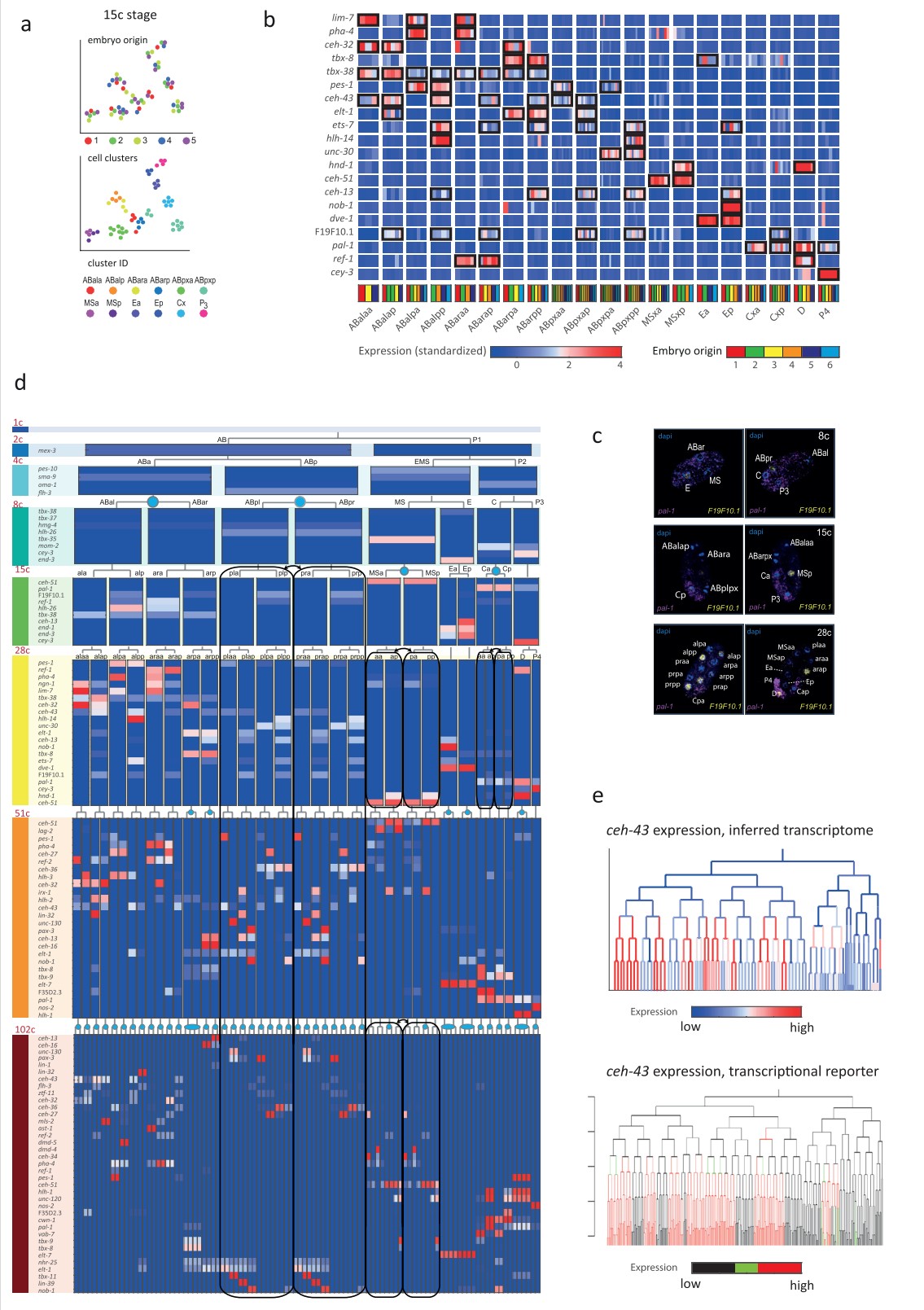

**Figure 2.** Inferring the transcriptomes of cell states throughout specification in the *C. elegans* embryo. (**a**) t-Distributed stochastic neighbor embedding (tSNE) clusters of cells from 15-cell stage embryos, shown in terms of their embryo of origin (top), cell clustering (middle), and inferred cell identity (bottom). (**b**) Gene expression for the indicated genes across the assigned cell states of the 28-cell stage. Rows correspond to genes and each bar within a cell state indicates the expression level of a sample. Black boxes indicate differential gene expression. Colors in the bottom row indicate the

*Figure 2 continued on next page*

*Figure 2 continued*

embryo of origin. (**c**) Double single-molecule RNA in situ hybridization showing spatial identity of *F19F10.1* expressing nuclei (yellow). The positions of all nuclei are shown with DAPI staining (blue), and the embryos were oriented with the posterior landmark gene *pal-1* (purple) (*Hunter and Kenyon, 1996*). Cell identities were manually annotated based on previous descriptions (*Supplementary file 4*). (**d**) Average expression of cluster identity genes through the first 8 cell divisions of embryonic development. Each bar represents the inferred transcriptome of a cell state and shows the standardized expression for the indicated genes, using the same color bar as in (**b**). Equivalent transcriptomes derived from sister cells are indicated by blue circles. Lineage symmetry gives rise to left/right equivalence groups indicated with black boxes. (**e**) Expression of *ceh-43* according to lineage of the inferred transcriptomes (top) and a transcriptional reporter strain (bottom), where highest expression is shown in red and no detected expression is black (see also *Figure 2—figure supplement 9*).

The online version of this article includes the following figure supplement(s) for figure 2:

**Figure supplement 1.** Sample collection and initial analysis.

**Figure supplement 2.** Cell state assignment and robustness analysis.

**Figure supplement 3.** Cell state assignment and robustness analysis.

**Figure supplement 4.** Cell state assignment and robustness analysis.

**Figure supplement 5.** Cell state assignment and robustness analysis.

**Figure supplement 6.** Cell state assignment and robustness analysis.

**Figure supplement 7.** Cell state assignment and robustness analysis.

**Figure supplement 8.** Number of cell states detected at each of the studied stages.

**Figure supplement 9.** Inferred and validated gene expression.

other stages). We required a new marker to distinguish some of the cell identities. Studying *F19F10.1* by single-molecule fluorescent in situ hybridization (FISH), we found that this previously uncharacterized ets-factor is expressed in the posterior daughter cell through 2-cell cycle divisions (*Figure 2c*). We further tested for the robustness of the cell identity assignments and ensured that each cell is most similar to its assigned identity (computed with itself excluded; *Figure 2—figure supplements 2–7*, bottom of each subfigure). We identified 5433 genes as being differentially expressed within a stage, collectively across all of the stages (see Methods). Of these, 395 are TFs during this period of cell specification (*Supplementary file 6*).

For some cell identities within a stage we were not able to distinguish distinct transcriptomes, and we thus denoted these as composing 'equivalence groups'. These may either be attributed to daughter cells whose transcriptomes have yet to diverge, such as cells Ca and Cp in the 15-cell state, or to sets of cousins whose sub-lineages form repeating units, such as cells ABplp and ABprp (also of the 15-cell stage) (*Sulston et al., 1983*). In the first case, this indicates that sister cells are transcriptomically very similar shortly after division. Altogether we describe 119-cell identities with distinct transcriptome profiles, which by stage, are as follows: 1 (1-cell), 2 (2-cell), 4 (4-cell), 6 (8-cell), 11 (15-cell), 20 (28-cell), 38 (51-cell), and 37 (102-cell), with 64 of these representing equivalence groups of at least two cell states (*Figure 2d*, *Supplementary file 5*, and *Figure 2—figure supplement 8*). For each cell state that we assigned, we have an average of seven replicate cells (*Supplementary file 4*). We validated our annotations by imaging GFP reporters and found good correspondence, as illustrated for *ceh-43*, *dmd-4,* and *unc-30* (*Figure 2e* and *Figure 2—figure supplement 9*). The delay between the GFP expression and the mRNA detected here is expected as previously described (*Murray et al., 2012*).

We next asked if the genes that are members of the same gene family have similar spatial and temporal expression arrangements. To test for this mode of expression, we considered groups of genes with the same DNA binding domain, as annotated by the PFAM database (*Finn et al., 2016*). For each cell, we queried for enrichments or depletions of genes of specific domain families, among its expressed TFs. As an example, the 'ABplppp' cell differentially expresses 14 TFs according to our analysis, including seven homeobox domain (HD) TFs, though overall HDs only account for 11% of the TFs (*Figure 3a*). We used the hypergeometric distribution to compute a significance of p=0.00015 for this overlap, given the overall number of TFs and the number of HD and 'ABplppp' TFs. Conversely, only two of the 69 TFs expressed by the Eala cell are HDs, a significant depletion (p=0.01, hypergeometric distribution). *Figure 3b* shows the overall pattern of enrichments and depletions of HDs across all of the examined cells, indicating enrichment of HDs in the 51-cell stage, in particular in the ABp

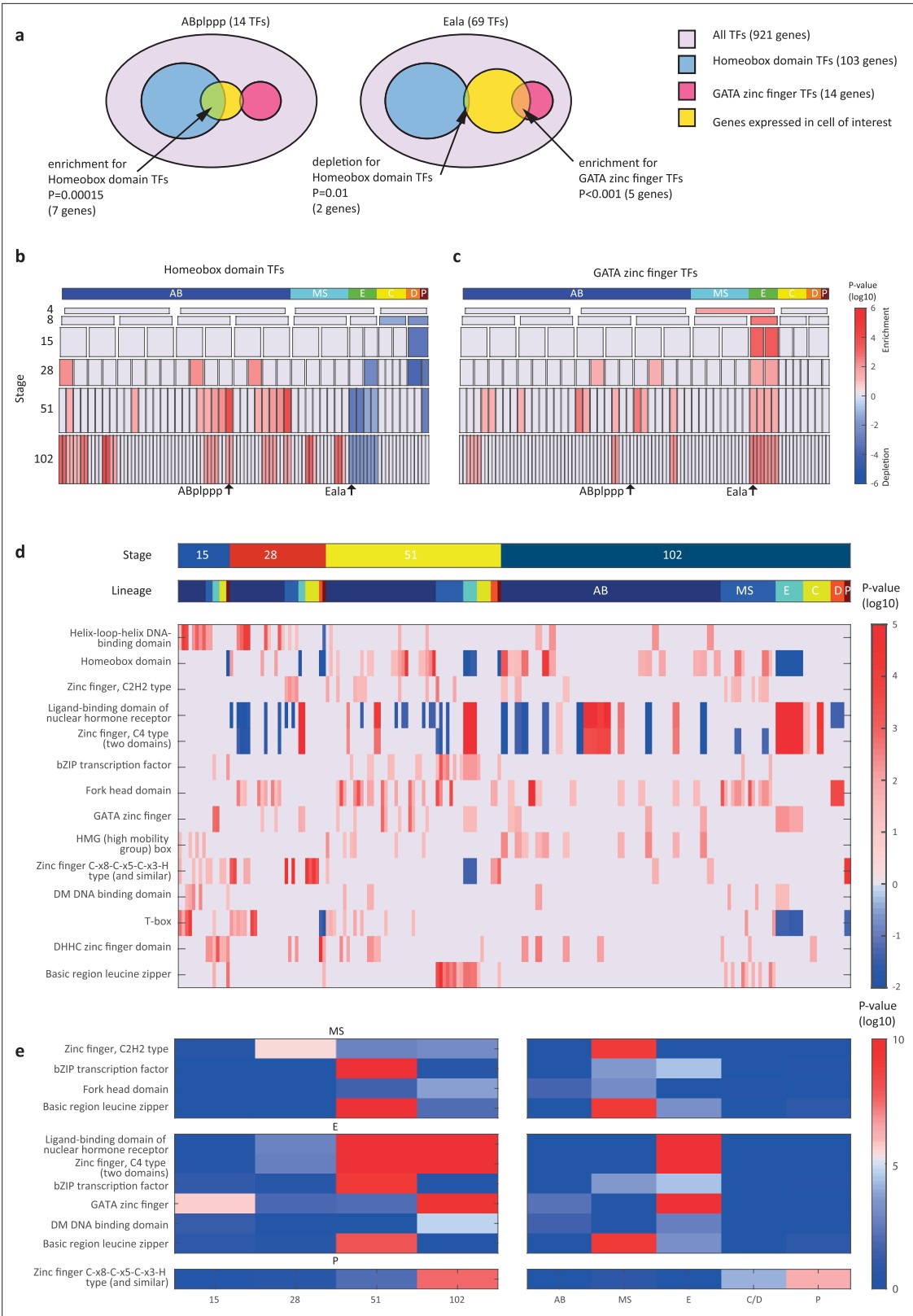

**Figure 3.** Transcription factor (TF) gene families have spatiotemporal specificities. (**a**) Detecting enrichment and depletion of genes of the same gene family, among the TFs expressed by a cell, using the hypergeometric distribution. (**b–c**) For the homeobox domain (**b**) and GATA zinc finger domain (**c**) gene families, the enrichments and depletions are indicated for each cell of the lineage, plotted as in *Figure 2d*. (**d**) TF family stage and lineage enrichments. For each cell (column), the enrichment and depletion for the genes of a TF family (row) is indicated (similar to **b, c**). The top bars indicate

*Figure 3 continued on next page*

*Figure 3 continued*

the lineage and stage of the cell. (**e**) For the indicated TF family the significance for enrichments is shown using the t-test between the values shown in (**d**) for the noted stage and lineage.

lineage, and depletion in the E lineage. For the GATA type zinc factors we found enrichment in the E lineage, as well as some of the AB cells (*Figure 3c*).

Examining the pattern of such biases, we found that TF families showed significant enrichments across lineages and developmental stages (*Figure 3d–e*). Four TF gene families are enriched in the AB lineage: the T-box domain, the helix-loop-helix (HLH) DNA-binding domain, the HD domain, and the high mobility group (HMG) box domain (*Figure 3d*). Each of these families also has enrichments for specific AB stages, from early expression (T-box and HLH) to later expression (HD and HMG).

From *Figure 3* analysis we concluded that the homeodomain (HD) TF family in particular exhibited lineage-specific expression. Studying this in more detail, we observed that in the 28-cell stage, each of the 16 AB cells contains expression of at least one homeodomain gene, with 8 distinct cell state patterns (*Figure 4a*). This is mostly also apparent in the 51- and 102-cell stages. The expression patterns of all homeodomain genes across the lineage are not mono-phyletic. For example, *ceh-43* is

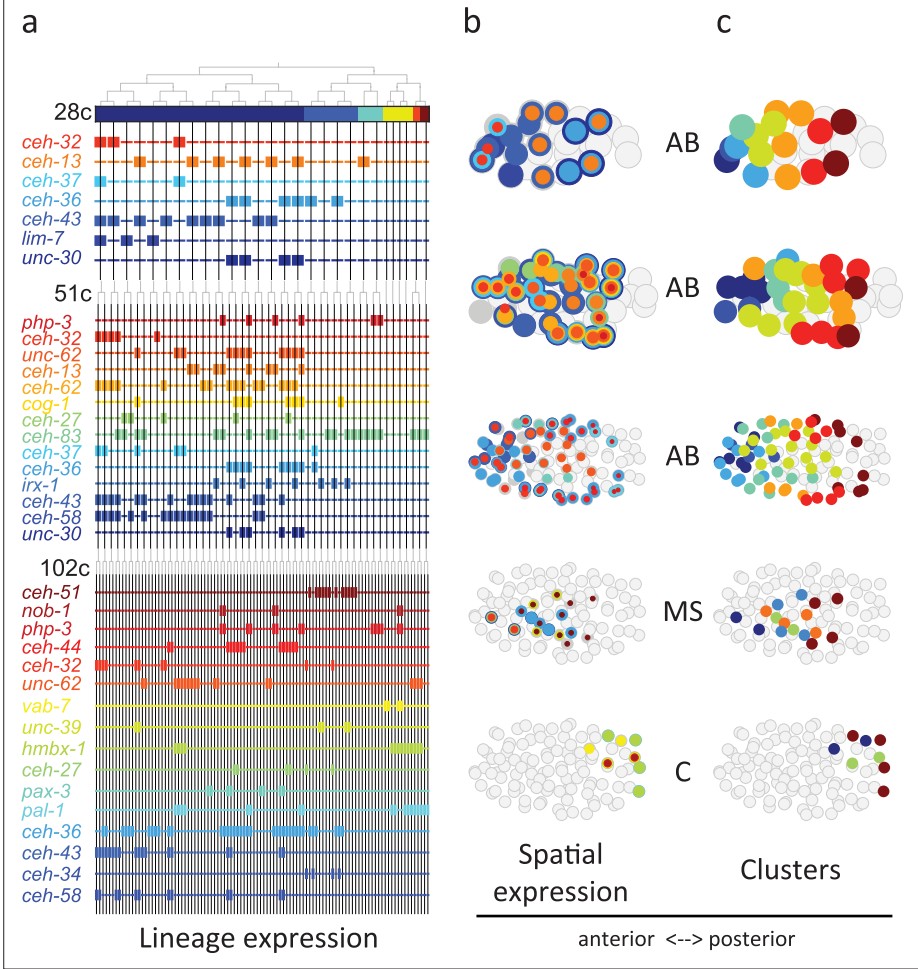

**Figure 4.** Homeodomain genes are expressed in lineage-specific stripes across the anterior-posterior axis. (**a**) Lineage expression patterns for the indicated genes in the 28-, 51-, and 102-cell stage. Thick and thin lines indicate binarized 'on' and 'off' expressions, respectively. (**b**) Three-dimensional expression patterns separated by lineage (AB/MS/C) for the genes indicated in (**a**). Circles represent cells in the embryo at the particular stage examined. Expression of each gene is indicated by a circle of a different size and color. Light gray cells indicate cells in other lineages. (**c**) Clustering of cells according to gene expression of the indicated lineage and stage. Colors distinguish K-means clustering.

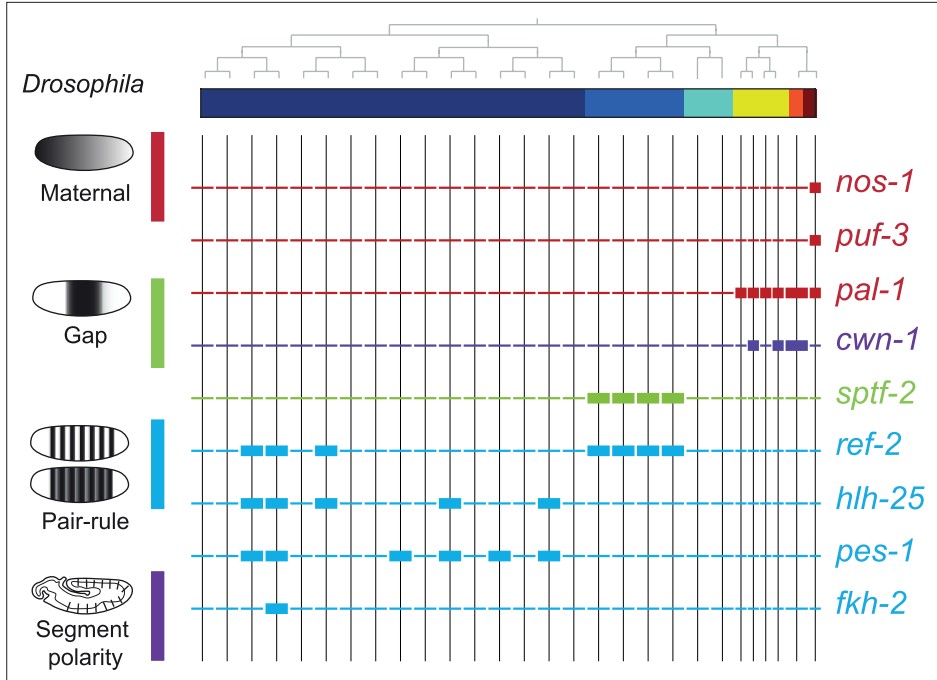

**Figure 5.** Expression of the orthologs of *Drosophila* specification genes in *C. elegans*. Lineage expression patterns in the 28-cell stage for the indicated genes whose orthologs function as maternal, gap, pair-rule, and segment polarity genes in *Drosophila*. Thick and thin lines indicate binarized 'on' and 'off' differential expressions, respectively.

expressed in nine of the sixteen AB cells at the 28-cell stage in a complex pattern (*Figure 4a*). Thus, continuous symmetry breaking (*Zacharias and Murray, 2016*) does not appear to be a mechanism by which differential expression is generally established.

We asked if the relative spatial location of the cells within the embryo can provide insight into the observed homeodomain expression patterns. For this analysis, we mapped the binarized expression of genes onto the known three-dimensional locations of cells in the embryo. Studying the 28-cell stage, we found that the homeodomain genes expressed in each lineage revealed dorsal-ventral stripes along the anterior-posterior axis (*Figure 4b*). For example, the *ceh-43*-expressing cells, deriving from distinct AB sub-lineages, are physically proximate. Clustering the AB lineage cells into eight groups according to their binarized expression, we mapped the gene expression combinations of each stripe. One stripe, for example, includes the expression of *unc-30*, *ceh-36*, and *ceh-13*; the adjacent stripe has expression of *unc-30* and *ceh-36* (*Figure 4b*). At the 51-cell stage, 4 new homeodomain genes are expressed in the AB lineage, and again position along the anterior-posterior axis correlates with gene expression. In the 102-cell stage, expression becomes more complicated as the AB lineage forms an external shell around the gastrulating non-AB cells. Generalizing this approach, we also found distinct positional systems for the MS and C lineages (*Figure 4b and c*). For the MS lineage, *ceh-32* and *ceh-51* are in the anterior and posterior, respectively, while for the C lineage *vab-7* (even-skipped) and *hmbx-1* are in the anterior and posterior, respectively. We thus provide evidence that distinct regionalization codes occur across the embryonic sub-lineages.

To compare the patterning code elucidated for *C. elegans* embryogenesis to the well-studied patterning pathway in *Drosophila*, we also studied the expression of *C. elegans* orthologous genes. *Drosophila* development proceeds through a series of molecular events which partition the syncytial embryo along the anterior-posterior axis (*Wolpert et al., 2019*). We found that the *C. elegans* genes such as *nos-1,2* (*nanos*) and *puf* genes (*pumilio*) which are maternal genes in both systems are restricted to the primary germ cell lineage in *C. elegans* (*Figure 5*). Moreover, gap and pair-rule genes, such as *pal-1* (*caudal*) and *cwn-1* (*wg*), show C lineage and sub-lineage restriction, respectively (*Figure 5*). Thus, it is evident that while *Drosophila* patterning follows in a lineage-independent

manner, in *C. elegans*, lineage dependency is coincident with specific expression of patterning genes across the founder cell lineages.

## Discussion

Collectively, our analysis reveals that during embryogenesis, in between the early signaling period – involving *Notch* (**Priess, 2005**) and *Wnt* (**Rocheleau et al., 1997**; **Thorpe et al., 1997**) signaling – and the later cell fate differentiation period – involving myogenic factors such as *myoD/hlh-1* (**Chen et al., 1992**) and *GATA/elt-2* (**Fukushige et al., 1998**) – there is a period of lineage-specific patterning. Three of the homeodomain genes whose expression we have described – *ceh-13*, *php-3*, and *nob-1* (**Figure 4**) – are members of the HOX sub-family, known to provide regionalization during embryogenesis in bilaterians (**Carroll et al., 2004**). Indeed, we observed that expression of these HOX genes transcends the founder lineages and patterns the posterior region in multiple lineages (**Figure 4**). However, we find 51 other homeodomain genes that likely pattern the lineages and set up locations in the embryo. This is a large genetic repertoire relative to *Drosophila* (97 genes in *C. elegans*, relative to 104 in *Drosophila*) despite the higher complexity of the latter in terms of cell types and total number of cells. Thus, in species where regionalization is under lineage control, as opposed to morphogen gradients, there is apparently a greater necessity for distinct positional systems across distinct sublineages – here the founder cell lineages. We find lineage-specific gene expression stripes (**Figure 4**), suggesting that such a developmental patterning mechanism may not be restricted to segmented organisms but is rather a universal feature of animal embryogenesis. The centrality of homeodomain genes to this system highlights the adaptability of this TF family to both lineage-dependent and position-dependent functions. It will be interesting to further compare other species at the single-cell level in order to trace the evolutionary origin of gene expression developmental patterning codes.

## Methods
### Cell isolations

Embryos were isolated from adult worms and removed from their egg shells as described in **Edgar and Goldstein, 2012**. Individual embryos were manually dissociated by gentle pipetting through progressively smaller glass-bore pipettes in calcium/magnesium-free EGM. Cell collection was limited to a 30 min window from the beginning of the dissociation process. For embryos with more than 28 cells, collection was performed as a team. Cell losses during dissociation were noted and single dissociated cells from an individual embryo were counted, collected as previously described (**Hashimshony et al., 2012**), and stored at –80°C until further processing.

### Single-cell RNA-Seq

Individual embryo, multiplexed single-cell RNA-Seq libraries were produced following the CEL-Seq2 protocol (**Hashimshony et al., 2016**). Briefly, individually harvested cells were barcoded with a reverse transcription reaction, and RNA molecules were tagged with unique molecular identifiers (UMIs) (**Kivioja et al., 2011**). All barcoded cells from a single embryo were then combined into a single IVT amplification reaction which was used to generate an Illumina library for sequencing. Multiple sequencing libraries with compatible Illumina sequencing indexes were sequenced together on an Illumina HiSeq 2500 using standard paired-end sequencing protocols. For Read 1, used to determine the barcode, the first 15 bp were sequenced and for Read 2, used to determine the identity of the transcript, the first 35 bp were sequenced. The CEL-Seq2 pipeline is available at GitHub (copy archived at **YanaiLab, 2017**). Mapping of the reads used BWA33, version 0.6.1, against the *C. elegans* WBCel215 genome (bwa aln -n 0.04 -o 1 -e -1 -d 16 -i 5 -k 2 -M 3 -O 11 -E 4). Read counting used htseq-count version 0.5.3p1 defaults, against WS230 annotation exons.

### Initial processing of the CEL-Seq2 data

Gene expression values were calculated as UMI-corrected reads normalized to the number of transcripts per 50,000; only cells with more than 10,000 mapped reads were included in the analysis. In order to compare gene expression patterns between embryos, gene expression was first standardized among the cells of an individual embryo (such that the mean expression for each gene across the cells

of the embryo is 0 and the standard deviation is 1), and these standardized expression profiles were used for comparisons between embryos.

## Cell clustering

t-Distributed stochastic neighbor embedding (tSNE) (*van der Maaten and Hinton, 2008*) was performed on full dataset using the top 1000 most variable genes (according to the Fano factor), with 20 initial dimensions and perplexity set at 50. For reproducibility, a non-random initiation of the objective function was set (https://lvdmaaten.github.io/tsne/).

## Cell identity inference

For each of the eight examined stages – 1-cell, 2-cell, 4-cell, 8-cell, 15-cell, 28-cell, 51-cell, and 102-cell – the isolated cells were first clustered according to their position on a tSNE plot using genes of known expression patterns (*Supplementary file 2*). We next manually and iteratively accepted clusters based upon their homogenous embryo composition, by removing the accepted cluster and re-clustering the remaining cells. These clusterings were based upon the restricted set of genes with known cell identity (*Supplementary file 2*). Cell state was then annotated to the clusters by comparing the average gene expression of the cluster with a collated database of known gene expression patterns (*Supplementary file 3*). The robustness of the clustering was tested by only retaining cells whose expression is most similar to the mean expression of each cluster, computed without it (*Figure 2—figure supplements 2–7*). The cell state transcriptome was then computed as the average expression of all members, together with standard deviation (*Supplementary file 6*).

## FISH analysis

Custom Stellaris FISH probe sets were designed against the entire coding sequence of *F19F10.1* (25 non-overlapping probes labeled with CAL Fluor Orange 560 Dye) and pal-1 (25 non-overlapping probes labeled with C3-Fluorescein Dye), using the Stellaris FISH Probe Designer (Biosearch Technologies, Inc, Petaluma, CA, USA). FISH was performed according to the protocol available on the Stellaris website for fixation of embryonic *C. elegans* material, with the following modification: all steps were performed in 1.5 mL Eppendorf tubes. Hybridization of two non-overlapping probe sets was performed using 125 nM of each probe set contemporaneously. Within 48 hr from completing the hybridization protocol, embryos were mounted on glass slides under #0 coverglass (85–115 µM thickness), supported on each corner by small amounts of plasticine to avoid crushing the embryos. Slides were imaged immediately after mounting with a Zeiss LSM 510 confocal microscope.

## Lineage tracing and single-cell fluorescence quantification

Embryo mounting and live imaging were performed as previously described. Briefly, 2- to 4-cell stage embryos were mounted between two cover slips in M9 buffer containing 20 µm microspheres (Polyscience). Dual-color 3D time-lapse imaging was performed using a Spinning disc confocal microscopy. Images were acquired with 30 focal planes at 1 µm Z resolution for every 75 s during a 6 hr duration of embryogenesis. Ubiquitously expressed mCherry was used for automated cell identification and lineage tracing using StarryNite program followed by systematic manual correction of errors (*Santella et al., 2014*). Protein fusion GFP was used to quantify expression level of corresponding gene in each cell using a previous established method (*Murray et al., 2008*). GFP intensity was measured by calculating the average GFP intensity for each cell and then with the local background intensity subtracted. GFP intensity at different time point for a same cell was averaged to quantify expression level of a gene in a cell. These data are further described in the recently published atlas (*Ma et al., 2021*).

## Differential gene expression analysis

For each gene, at each stage, we identified differential gene expression using the following approach. We compared the standardized gene expression values across the cell states of the stage, and identified the two groups of cell states that correspond to the most significantly different groups, assayed by a t-test p-value. In order to establish the significance, we then asked if that p-value is better than that expected by shuffling the expression values across states. To find the two groupings with the best p-value, we began by computing a t-test between the expression values for each cell state and the rest of the expression levels of the remaining states. The cell state with the best p-value was used to

seed a group and we next computed the p-value of a super-group composed of this cell state in turn with each other cell state, comparing with the expression values of the remaining cell states. If one of these super-groups had a better p-value than the original best p-value state alone, we selected it and continued this process, iteratively adding to the super-group until the p-value can no longer improve. This process results in a 1-cell state or a super-group of cell states with expression and a group without expression. Each gene will thus have a binary profile and an associated p-value. To select the genes with sufficiently significant p-values we repeated the procedure on 10,000 shuffled sample to cell state assignments. Sorting the p-values from the shufflings, the p-value at 1% of this list was used as the threshold, for a false discovery rate of 1%. We excluded differential expression of genes expressed in more than 50% of the cells of a given stage, and also restricted differential expression to those genes whose maximum expression is at least three quarters of a $\log_{10}$ TPM unit greater than the minimum expression at the given stage.

## Enrichments and depletions of TF families in cells

For each cell, we computed both the enrichment and depletion of the set of genes in a given TF family among its expressed TFs. The enrichment and depletion is computed as the cumulative hypergeometric p-value with and without subtracted from unity, respectively. The smallest number between these two tests distinguishes an enrichment from a depletion (see *Figure 3a*), and we multiplied the $-\log_{10}$ p-value by 1 and $-1$, to distinguish these in the plots shown in *Figure 3b, c*. To compute the significance of enrichments for particular lineages and stages, we used the t-test on the $-\log_{10}$ p-values of the enrichment values.

## Materials availability statement

The complete dataset has been deposited to the NCBI GEO database GSE83523. The authors declare no competing financial interests. Correspondence and requests for materials should be addressed to IY (itai.yanai@nyulangone.org).

## Acknowledgements

We thank the Technion Genome Center. This work was supported by a European Research Council grant (EvoDevoPaths). We are grateful for the EPIC2 dataset, which we used extensively. We also thank Yun Yan for work on an early version of this project.

## Additional information

### Funding

| Funder | Grant reference number | Author |
| --- | --- | --- |
| European Research Council | EvoDevoPaths | Itai Yanai |

The funders had no role in study design, data collection and interpretation, or the decision to submit the work for publication.

### Author contributions

Alison G Cole, Conceptualization, Data curation, Investigation; Tamar Hashimshony, Conceptualization, Data curation, Visualization, Methodology; Zhuo Du, Resources, Data curation; Itai Yanai, Conceptualization, Formal analysis, Writing – original draft

### Author ORCIDs

Alison G Cole http://orcid.org/0000-0002-7515-7489
Tamar Hashimshony http://orcid.org/0000-0002-4786-838X
Zhuo Du https://orcid.org/0000-0002-6322-4656
Itai Yanai https://orcid.org/0000-0002-8438-2741

Reviewer #1 (Public Review): https://doi.org/10.7554/eLife.87099.3.sa1

Reviewer #2 (Public Review): https://doi.org/10.7554/eLife.87099.3.sa2
Reviewer #3 (Public Review): https://doi.org/10.7554/eLife.87099.3.sa3
Reviewer #4 (Public Review): https://doi.org/10.7554/eLife.87099.3.sa4
Author Response https://doi.org/10.7554/eLife.87099.3.sa5

## Additional files

### Supplementary files

• Supplementary file 1. Characteristics of the dataset. Rows correspond to a studied embryo. The average number of transcripts and genes detected are shown.

• Supplementary file 2. Gene markers from the literature. Cell state markers for the expression profiles used for assignment were curated from the literature (*da Veiga Beltrame et al., 2022*) and, in particular, one large study (*Murray et al., 2012*).

• Supplementary file 3. Expression of marker genes across cell identities. For each cell state the expression of the cell state markers used for annotation (*Supplementary file 2*) is indicated.

• Supplementary file 4. Inferred cell states. Each row corresponds to 1 of the 119 inferred cell states. The third column indicates the number of samples (scRNA-Seq cells) annotated to each cell state.

• Supplementary file 5. Cell state assignments. For each sample (scRNA-Seq), the table provides the GEO accession number, sample name, embryo ID, annotated cell state, and the number of transcripts detected.

• Supplementary file 6. Gene expression dataset. For each gene, the average, standardized, differential, and standard deviation of expression is provided (see Methods) across the 119 detected cell states.

• MDAR checklist

### Data availability

The complete dataset has been deposited to the NCBI GEO database GSE83523.

The following dataset was generated:

| Author(s) | Year | Dataset title | Dataset URL | Database and Identifier |
|---|---|---|---|---|
| Cole AG, Hashimshony T, Yanai I | 2023 | Transcriptome characterization of embryogenesis at a single cell resolution [*C. elegans*] | https://www.ncbi.nlm.nih.gov/geo/query/acc.cgi?acc=GSE83523 | NCBI Gene Expression Omnibus, GSE83523 |

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
