## [Editor Report · eLife assessment]

This **valuable** work fills a gap in the mapping of gene expression patterns in the early embryo of *C. elegans*. The presented data are **solid** and provides a resource for future analysis.

---

## [Referee Report · Reviewer #1 (Public Review)]

*C. elegans* is a pre-eminent model for developmental genetics, and its invariant lineage makes it possible in theory to define molecular features such as gene expression comprehensively and at single cell resolution across the organism.

Previously published single-cell RNA-seq studies have mapped gene expression across the lineage through the 16-cell stage (Tintori et al 2017, Hashimshony et al 2016), and at later stages (Packer et al 2019, with good coverage starting at the 100-cell stage and some coverage at the ~50-cell stage). This left the critical period around gastrulation (~28-cell and ~50-cell) without comprehensive transcriptome data. This study covers this gap with a heroic effort involving the manual isolation and analysis of over 800 cells from embryos of known stage, combined with painstaking curation using known markers from small scale studies and larger imaging-based expression atlases. Importantly, the dataset overlaps at early and late stages with data prior studies.

The data quality and overlap with Tintori and Packer datasets both appear high, but to make this inference required additional analysis from Supplemental Table 6 by this reviewer as it is not explored or described in the manuscript. Analyses demonstrating continuity with these datasets would greatly increase the value of the resource.

The authors show that specific lineages and stages preferentially express TFs with different classes of DNA binding domains. This extends previous work implicating homeodomains as preferentially involved in nervous system patterning and as enriched in neural and muscle progenitors in mid-stage embryos.

They also show that *C. elegans* homologs of *Drosophila* early embryonic regulators (which function based on spatial position in that system) tend to also be patterned in early *C. elegans* embryos, but with lineage-specific patterns. This conserved use of regulators would be fairly remarkable given the dramatically different developmental modes in these two species, although this observation is not backed up by quantitative analyses.

Finally, there is an argument that combinations of TFs expressed in lineage-specific patterns give rise to "stripe" patterns. This section is also not based on statistical analyses but suggests the possibility that lineage and positional regulation may be more convoluted than was previously thought.

---

## [Referee Report · Reviewer #2 (Public Review)]

The *C. elegans* embryo has been model system of study for more than 30 years because of the ease of doing forward and reverse genetics, coupled with its nearly invariant lineage which allows a description of development at high resolution. 4D time lapse imaging coupled with spatially resolved gene expression has enabled identification of transcriptional signatures of cells in space and time, and in the past decade this has been advanced with single-cell transcriptomics methods, using individually isolated embryonic cells (which can retain their identity) or by deconvolving complex mixtures of early cells. Recent work using these methods has resolved spatiotemporal expression patterns for many genes, defining cells up to gastrulation stage, but then changing to more tissue-specific patterns during morphogenesis. A key paradigm of specification in *C. elegans* and other systems is that early maternal factors initiate or restrict patterns of transcription factor expression from the zygotic genome. Combinatorial expression patterns and some symmetries broken by autonomous or extrinsic cell inductions ultimately program lineages towards their fates. To date, only simple networks have been elucidated, as the increasing complexity of these networks and the high level of redundancy has made functional dissection of such pathways difficult. Hence, almost all of the work in recent years has been descriptive.

In this work the authors fill a knowledge gap from the early embryo (~16 cells) to the ~100-cell stage and describe new patterns of gene expression. They reconcile their findings with that of others who have defined expression patterns using other methods, such as scRNA-Seq from complex mixtures of cells, and from transcription factor expression analyses. The resulting description of embryonic develop is the most precise to date, and offers a potentially useful resource for other researchers.

The authors attempt to use their results to find patterns of gene expression that could hint at phylogenetic conservation of specification mechanisms. They find some supporting evidence that expression of homeobox genes occurs in anterior-posterior stripes, which recalls the elaborate A/P patterning system elucidated in the *Drosophila* embryo, which belongs to the sister phylum Arthropoda in the Ecdysozoan clade of molting animals. It felt as if the authors chose the Hox genes they need to support this conclusion.

Some caveats exist to the work. The expression patterns seem to be well-validated, and following prior work from the Yanai group are likely to be strongly correlated with expression in living embryos. When cells are separated, they could lose some expression patterns that require cell-cell interactions, so it is expected that there might be a small minority of expression patterns that are more complex than what has been documented here.

A major caveat is the idea of the stripes of Hox expression. I just did not find these arguments to be compelling. Seeing these 'stripes' requires organizing the data in a way that maximizes their appearance, for one. Since there is not a lot of movement of cells away from their birth in the early embryo, the AB descendants are anterior to those of MS, anterior to those of E, anterior to those of C, D, and P4. Lineage-specific expression will just naturally fall into 'stripes'. Second, the conservation of Hox expression patterns typically comes with collinearity of the genes along the length of a chromosome (i.e. the so-called Hox clusters) with expression along the body axis, as well as posterior-to-anterior fate transformations when Hox specification is disrupted.

A minor note is the detection of an enrichment of GATA factors in the early E lineage. This has now been found to be a derived condition even within the genus see Broitman-Maduro et al. Development 149 (21): dev200984, as other species like C. angaria show only a simpler network of elt-3 -> elt-2. This suggests that many of the other patterns of gene expression, particularly in the early embryo, could be highly derived as well; some caution is warranted in generalizing the results as being conserved with arthropods as some of this could be convergent.

Given what the authors are proposing about Hox stripes, some omissions of prior work were surprising (or maybe I missed them). For example, a comprehensive study of Hox genes in C. elegans by Hench et al. (2015) (PLoS One 10(5): e0126947) evaluated all the homeobox genes and examined their genomic locations and expression patterns in the embryo at high spatiotemporal resolution. Work from the Hobert lab (Nature 2020, 584(7822):595-601) showed how homeobox codes specify classes of *C. elegans* neurons, and the Murray lab (PLoS Genet. 18(5):e1010187) examined Hox control of posterior lineage specification at high resolution, with functional assays.

The Discussion section of the paper is brief, consistent with the descriptive nature of the work overall, but it would have been nice to see the findings related to other published studies as indicated above.

---

## [Referee Report · Reviewer #3 (Public Review)]

The authors claim that this dataset covers a timepoint of embryogenesis that is not well covered in the other published single cell datasets (Tintori et al 2016 and Packer et al 2019). The Tintori data indeed do not cover the 28-102-cell stages sufficiently, but it is unclear how the data presented here compare to the Packer et al data. It is true that the Packer et al data have fewer cells at earlier timepoints than at later ones, but given that they sequenced tens of thousands of cells, they report that they still have ~10,000 cells <210 min of embryogenesis. It seems that if the authors want to make any claims about how their data enables exploration of a stage that was previously not accessible, this would require a better comparison to the available data.

The authors provide thorough support for how they assigned cell identities in their data. It is surprising though that at the 102-cell stage they only identify 37 unique cell identities. They suggest that this is because there are many equivalence groups at this stage. However, I would strongly encourage the authors to perform a similar analysis or otherwise compare their obtained identities with the data from Packer et al. 2019. It seems possible that given the low number of cells in this dataset, the authors are missing certain identities and it would be important to know this.

The main analysis the authors perform is to look at expression patterns of various classes of TFs and ask whether they are enriched in particular lineages or at specific timepoints. This analysis is interesting but would be more informative if the authors provided in Figure 3d the numbers of each class of TFs. The authors then focus on the homeodomain class of TFs as they display interesting lineage-specific expression patterns, which when mapped on the embryo form stripes. The stripe pattern however is not that obvious, at least not as shown in Figure 4b. Perhaps separate embryo schematics showing the different TF expression patterns would show this more clearly. Moreover, given the relatively small number of cell identities found in this dataset (particularly at the 102-cell stage), a similar analysis using the Packer data would provide further support to these patterns. The localization of cells with shared expression patterns does show a stripe pattern at the 28-cell stage, but also not so clearly beyond this timepoint.

I am also unsure about the validity/value of the comparison of the stripes to *Drosophila* and the centrality of homeodomain TFs to anterior-posterior positional identity. First, it would be important to map other TFs, very likely there are several other TFs that correlate with positional identity. Also, even if the expression of the homeodomain TFs in *C. elegans* form stripes, there are still several cells within that stripe that do not express these TFs, it is thus unclear whether these TFs encode positional information or the identity of cells with different positions in the embryo.

---

## [Referee Report · Reviewer #4 (Public Review)]

This is an admirable piece of work. The authors build on a previous dataset they assembled, but expand it to include all stages of early development in the nematode *Caenorhabditis elegans*. Cell collection was done manually, which is very impressive, and is clearly far better than pooled unidentified cells. I will not comment on the specific sequencing and analysis, since this is not my expertise, but will comment on the general conclusions and comparative framework in which the authors place their results.

While the Introduction and Discussion sections are actually fairly short, much of the presentation of the results is based on a certain comparative framework, which is explicitly a comparison between *C. elegans* and *Drosophila melanogaster*. This is an important perspective, but I feel the authors' interpretation is in some places exaggerated and in other places almost trivial.

*Drosophila* and *C. elegans* are two of the main models for developmental biology. However, it has been clear for over two decades that both species are highly derived and specialized and therefore, treating them as representative for their taxa is problematic. Much of the authors' discussion hinges on the question of comparing syncytial and lineage-dependent development. The syncytial early development of *Drosophila* is very specific and is clearly a recent innovation within a restricted group of flies. The canonical *Drosophila* segmentation cascade is mostly a novelty and most elements within the cascade are recent. Specifically, the expression of gap genes in regional stripes is not found very broadly. Conversely, the polarizing role of Caudal is very ancient and is probably found in all Bilateria. When making comparisons with a distantly related species, it is important to keep this in mind. Not as much is known about development of other nematodes, but the little that is known indicates that *C. elegans* is also unusual, and specifically the eutelic development (conserved cell lineages in development) is not found in all nematodes.

The authors suggest that regional expression of transcription factors in stripes is a conserved characteristic of development. This is true for Hox genes and has been known for decades. The regional expression they show for other genes is not convincing as "stripes". It is no surprise that developmental transcription factors are regionalized, but linking this to the stripes of *Drosophila* gap genes and even more so to *Drosophila* pair-rule and segment-polarity genes is a bit far-fetched. Yes, many genes are expressed in restricted domains along the A-P axis, but that is all that can be said based on the data. Calling them "*Drosophila*-like" is unfounded.

---

## [Author Response]

The following is the authors’ response to the original reviews.

**Reviewer #1 (Recommendations For The Authors):**
1. In general given several of the "equivalence groups" were distinguished from each other in Packer et al's annotation, can the authors comment more on why they aren't able to distinguish them? Are the markers listed for those cell states in Packer not expressed appropriately in these data? Or are they expressed but the states are not different enough to form discrete clusters? I suggest the possibility that the analysis choices of 20 "initial dimensions" or 1000 most variable genes filtered out some of these differences which may be encoded in later principle components, or that the use of t-SNE projection is not sufficient to resolve these distinct states.1. I was a bit confused by the spatial gene expression analysis. Several distinct ideas appear to be posed in the text. These ideas aren't really supported by any quantitative analysis, just the visual patterns in Figure 4B/C which I'm not sure I always agree with.For example, ceh-43 expression is mentioned as having "physically proximate" expression. But it is well established that different lineages form specific spatial territories (e.g. Schnabel et al 1997). Thus it seems logical that genes with specific lineage patterns will have specific spatial patterns as well. If the claim is that the observed patterns are more clustered along the A-P axis than expected by chance given their lineal complexity then I'm not sure this is shown. Maybe some comparison with control lineage patterns of similar complexity of non-TFs or non-HD TFs could get whether these genes specifically are more spatially patterned? Visually it looks to me like some patterns are more like "blobs" or even lateral or D-V specific patterns than they are like "stripes."In addition there is a long history in the literature discussing the origin of position-specific patterns in *C. elegans* - most I'm aware of support the idea that positional information arises primarily from intrinsic lineage mechanisms (e.g. Cowing and Kenyon 1996). Perhaps the authors are making this same argument here, but if so this isn't clear from the text.Or maybe the authors are trying to make the argument that combinations of TFs encode more precise position than individual TFs? This seems more likely to me from the images presented still not well-supported without quantitative or statistical analyses.1. The comparison with Drosophila is interesting but also under-developed. I think all I would feel comfortable claiming from the data as shown is that genes that are spatially patterned in early fly development are also usually patterned in the *C. elegans* lineage. But to even say this is an enrichment over expectation would require more analysis.Minor comments:Methods: some statement about temperature control during cell isolation would be useful. In other words were embryos continuing to develop or put at low temperature such as in a cold room to prevent temporal differences between the first and last cells collected from a given embryo?Current links to data at GEO are incorrect and link to Levin et al 2016 instead. I was not able to access the raw single cell data, just the processed data in Table S6.The standardization of expression in embryos isn't well explained - would be good to expand a little on the types of batch effects being addressed and how this approach was chosen or a relevant citation.Page 2: Including P0 and cell deaths there are 1,341 branches in the hermaphrodite lineage (2n-1 for 671 terminal cells including deaths).-"as their each have" (grammar error)-"very large nuclear hormone receptor domain" (add "family")Page 3: As noted Packer et al largely missed cells prior to the 50-cell stage as described - but the reason for this is likely that the use of 10 micron filters or centrifugation to remove undissociated embryos also removes early stage cells.-"few new expressions occur" (grammar). Also, in both Tintori and Hashimshony datasets there well over 1000 newly expressed genes detectable (see for example Sivaramakrishnan et al 2021 biorxiv).Figure S1 would be easier to interpret with a legend explaining what fates are represented by each colorSome genes listed as markers in Figure S2 are not included in the marker table such as flh-3, oma-2, sma-9."New markers were required" - this is plural but only F19F10.1 is mentioned. Were other markers examined this way or should it be singular?In Figure S2 the lower ("robustness") plots are nice but could be explained more clearly. What is the nature of the "cell similarity score"? How many (if any) cells were excluded due to not being most similar to their own cluster?"transcriptomically very similar shortly after division" - can the authors comment on any information they have about how long after division the cells were collected?GFP reporter lineaging - the methods are minimally described (what brand of microscope, which strains/transgene/CRISPR configurations etc). And data are not presented. If these embryos are all incorporated into Ma et al 2021, that is fine, but should be clearly cited. Otherwise it is important in my view to include some way to access the quantitative values from the lineaging and understand these details."as illustrated for ceh-43, dmd-4 and unc-30" - were there other examples as suggested from this wording? I'd also note that similar fluorescent reporter imaging data have been published previously for all three genes listed (Walton et al 2015 for UNC-30, Ma et al 2021 for DMD-4 and CEH-43 protein reporters, Murray et al 2012 for dmd-4 and ceh-43 promoter reporters).Zacharias and Murray are cited as promoting "continuous symmetry breaking" but actually that review argued for a "non-monophyletic" architecture similar to that supported by the data .The text and figure don't always agree. For example mec-3 expression is listed in the text as part of one of the stripes, but mec-3 is not labeled on the figures.The stage of each embryo in figure 4B/C should be explicitly labeled (and maybe also given specific figure panel designations to clarify what statements in the text correspond to which figures).In the discussion it is unclear what the numbers "97 to 104" refer toThe scRNA-seq reads were mapped to a relatively old genome build and annotation set (WS230) - thus current users may find discrepancies with current gene names in WormBase. Also, since the CEL-seq data are 3' biased, it is worth noting that Packer et al found that a substantial number of genes (~1000) in a slightly later annotation set (WS260) were undercounted (sometimes dramatically) with the similarly biased 10x data due to incomplete 3'UTR annotations. While I would be reluctant to ask for a requantification for the purposes of the manuscript given the challenges of repeating the various analyses, it is worth explicitly mentioning whether this was dealt with.
**Reviewer #2 (Recommendations For The Authors):**
The writing was otherwise good, at least to my eye, and the data was presented very well and made freely available to other researchers. I am not as well-versed in the statistical methods and will leave comments on these to a better-equipped reviewer(s).Fig. 1 legend 'P' should be P4 (subscript 4).p. 9 'ceh-51' should be italicized.Only one factor seems to have been confirmed by smFISH, F19E10.1. There are available reporters, did they show a similar pattern? From CGC website: RW12347F19F10.1(st12347[F19F10.1::TY1::EGFP::3xFLAG]) V endogenous tagged reporter; RW11620 unc-119(tm4063) III; stIs11620 [F19F10.1::H1-wCherry + unc-119(+)] array reporter.
**Reviewer #3 (Recommendations For The Authors):**
Typo: on page 11, where it says nanog it should read nanos.
**Reviewer #4 (Recommendations For The Authors):**
I found some sentences and paragraphs to be a bit unclear. There are no page or line numbers in the manuscript, so I point in the general direction, and hope the authors find what I am referring to.2nd paragraph of the Introduction - "their" should be "they", but the sentence as a whole is not clear.3rd para. of the Intro. - The last sentence of this paragraph doesn't make sense. Please rephrase and/or break up into shorter sentences.1st Para. of Results - "the maternal deposit" is not clear. Perhaps "maternally deposited transcripts" or something similar.1st Para. after Figure 3. The last sentence "Thus, continuous symmetry breaking..." is unclear. What is "continuous symmetry breaking"? Please define and expand.Fig. 4 - the genes seem to be listed from posterior to anterior. The common way of presenting Hox gene lists and other regionally expressed genes is from anterior to posterior.For the benefit of the non-*C. elegans* crowd, please give names of Drosophila homologs where relevant (e.g., when comparing to Drosophila expression patterns)In a few places there are citations of popular science books or general textbooks (e.g., Carrol et al., 2004; Wolpert et al., 2019) . I think it would be better to cite review papers from the scientific literature or relevant primary papers.

We thank the four reviewers for their thoughtful and constructive comments. In light of their comments we made important revisions that are integrated in the final version, including also the correction of several typographical errors.